# Exploring the associations between number of children, multi-partner fertility and risk of obesity at midlife: Findings from the 1970 British Cohort Study (BCS70)

Sebastian Stannard[1,2,3]*, Ann Berrington[1,2]°, Nisreen A. Alwan[3,4,5]°

1 Department of Social Statistics and Demography, University of Southampton, Southampton, United Kingdom, 2 ESRC Centre for Population Change, University of Southampton, Southampton, United Kingdom, 3 Faculty of Medicine, School of Primary Care and Population Sciences, University of Southampton, Southampton, United Kingdom, 4 NIHR Southampton Biomedical Research Centre, University of Southampton and University Hospital Southampton NHS Foundation Trust, Southampton, United Kingdom, 5 NIHR Applied Research Collaboration Wessex, Southampton, United Kingdom

° These authors contributed equally to this work.
* S.J.Stannard@soton.ac.uk

**Data Availability Statement:** The BCS70 datasets generated and analysed in the current study are publicly available in the UK Data Archive repository

## Abstract

### Background

Early parenthood, high parity, and partnership separation are associated with obesity. However, the emergence of non-marital partnerships, serial partnering and childbearing across unions, means that it is important to consider their association to obesity. This paper examined the associations between number of biological children and multi-partner fertility (MPF)—defined as having biological children with more than one partner, with obesity at midlife.

### Method

The sample consisted of 2940 fathers and 3369 mothers in the 1970 British Cohort Study. The outcome was obesity (BMI 30 or over) at age 46. Fertility and partnership histories ascertained the number of live biological children and MPF status by age 42. The associations were tested using logistic regression adjusting for confounders at birth, age 10 and age 16. Adult factors recorded at age 42 including age at first birth, smoking status, alcohol dependency, educational attainment and housing tenure were considered as mediators.

### Results

For fathers, obesity odds did not differ according to number of children or MPF. In unadjusted models, mothers with one child (OR 1.24 95%CI 1.01–1.51), mothers who had two children with two partners (OR 1.45 95%CI 1.05–1.99), and mothers who had three or more children with two or more partners (OR 1.51 95%CI 1.18–1.93) had higher odds of obesity. In adjusted models, there remained an association between mothers with one child and odds of obesity (OR 1.30 95%CI 1.05–1.60). All other associations were attenuated when confounders were included.

(available here: http://www.cls.ioe.ac.uk/page.aspx?&sitesectionid=795).

**Funding:** This work was partly funded by the Economic and Social Research Council (UK) [ES/P000673/1]. The funders had no role in study design, data collection and analysis, decision to publish, or preparation of the manuscript.

**Competing interests:** The authors have declared that no competing interests exist.

## Conclusions

Mothers who had children with multiple partners had higher odds of obesity. However this association was completely attenuated when parental and child confounders were accounted for; suggesting that this association may be explained by confounding. Mothers who had one child only may be at increased odds of obesity, however this could be due to multiple factors including age at first birth.

## Background

Obesity is a major public health concern with estimates suggesting that one in four adults in the UK have obesity [1]. Obesity is associated with multiple morbidities including Type 2 diabetes, heart disease, strokes and some cancers including breast and bowel cancer [2–4]. Understanding the life course determinants of obesity is increasingly recognised; past events influence current health status and the accrual of events may lead to the accumulation of stressors that have long-lasting implications for health [5–7]. Partnership formation and dissolution, and childbearing represent major life events throughout most people lives. Scholars have shown that high parity [8–10], early parenthood [11,12] and partnership separation [13] are associated to obesity risk. However, given increasing levels of multiple partnership dissolutions, re-partnering, and blended families [14,15], it is important research considers the role of childbearing across multiple partnerships on obesity risk. This study considers a composite partnership and fertility variable that incorporates both number of biological children and multi-partner fertility (MPF), defined as having biological children with multiple partners, to explore whether these complex family dynamics are associated to obesity risk at midlife.

MPF is associated with several interrelated processes including early age at first birth, a higher number of biological children, multiple partnership dissolutions, serial partnering and lone parenthood. The instability and change resulting from MPF may lead to disruption and uncertainty across several domains including disrupted routines, changes in geographical location, custody arrangements of a child, becoming a single parent and losses of economic and social capital [16,17]. All of these life course processes have the propensity to increase stress, which can lead to sustained increases in blood pressure, hypertension, vascular hypertrophy [18,19], plaque formation [20] and an exacerbation of pathophysiology and symptomatology [20,21]. Stress can lead to chronic elevation of glucocorticoids, which can impede the action of insulin to promote glucose uptake [21,22] and can lead to poor health related behaviour including excessive food consumption, alcohol consumption, smoking and substance abuse [20–22]. Chronically elevated glucocorticoids, a result of stress, can impede the action of insulin to promote glucose uptake and promote the deposits of body fat leading to increased BMI [23,24].

MPF also encompasses early childbearing and high parity that may lead to permanent physiological alterations that increases the risk of chronic disease and poor physical functioning such as the Short Physical Performance Battery scale [25]. An early age at first birth can lead to lower oestrogen exposure and potentially harmful changes to lipid and glucose metabolism during pregnancy [10]. Women who give birth in their teens have a higher risk of developing eclampsia, pregnancy-related hypertension, lasting insulin resistance, osteoporosis and altered cholesterol profiles [26,27], and some of these conditions have a complex interrelated relationship with obesity [28–30]. Men, but especially women who experience early parenthood may experience disrupted educational and career progression that may increase the risk of

partnership breakdown and socioeconomic disadvantage [31], two factors that are associated with increased risk of obesity [32]. In contrast, those who enter parenthood when older have some psychological advantages and are often more resilient [33], given that they are more likely to have higher psychological hardiness [34] and have improved socioeconomic factors [33]. Finally, scholars have posited that the physical and caloric demand of repeated pregnancies can be determinantal for maternal outcomes and parenting stress [35–37], given that with an increasing number of children there is a growing need to manage time, resources and parental-child relationships.

The association between MPF and health in adulthood is likely to differ by gender. The biological consequences of pregnancy and reproductive factors such as age at menarche, the menopause, and breastfeeding [38–40] are only relevant for women, and the social implications of MPF are different for mothers and fathers. Firstly, fathers are less likely to retain custody of children following a parental separation [41], and this has been found to lead to poorer health-related behaviour, poorer self-reported health and being less happy as a parent [42]. For mothers, MPF may expand kin networks, whereas for fathers, kin networks may not expand [43]. Fathers have been found to struggle to negotiate obligations of times and resources to multiple groups of children [16]. While a mother may experience elevated stress managing her current residential family, her relationship with the biological parent of her first child and her children's relationship with her new coresidential partner. MPF may also lead to resource 'swapping' where resources are 'swapped' from a child in a previous relationship to current residential children [44,45]. 'Swapping' may be particularly pertinent for mothers who are more exposed to this loss of social and economic resources previously provided by a child's father. Given these differences, it is important to consider mothers and fathers separately in the association between MPF and obesity.

Adjusting for confounders to account for selection, we explore the associations between the number of biological children and MPF and the odds of obesity at midlife, for mothers and fathers. We hypothesis that any observed relationships may operate through both biological and social/environmental pathways.

## Data and methods

### Data

We utilise data from the birth, age 10, age 16, age 30, age 42, and age 46 sweeps of the 1970 British Cohort Study (BCS70) which has followed 17,196 participants from across England, Scotland and Wales born in a single week of 1970 [46]. A cohort profile providing information about the background to the study and full survey methods for each sweep is reported elsewhere [47]. Ethical approval for the cohort was granted by the National Health Service Research Ethics Committee, and all participants provided fully informed consent. Ethics approval for this study was also granted by the University of Southampton Ethics Committee (Reference number: 41778).

**Sample.** The analytical sample includes all cohort members who had measured obesity at age 46 (n = 6301).

**Exposure assessment.** Number of biological children is defined as the total number of live births (excluding miscarriages and/or abortions) the cohort member had had prior to age 42 (categorised into 1, 2, 3+ children). Whether or not the respondent had experienced MPF is estimated by comparing the birth month and year of each biological child to each coresidential union start and end months. Two assumptions are made: firstly, those births occurring up to six months prior to union formation were linked to this new union because non-coresidential relationships either dissolve or transition to coresidential relationships (either marriage or

cohabitation) fairly quickly following conception [48,49]. Secondly, births up to nine months following a union dissolution are linked with the previous partner as a child would have been conceived at the time of the previous union. The MPF variable is derived into a binary measure (yes or no) and combined with the total number of biological children ever born/fathered to make a composite variable with categories: one child, two children with one partner, two children with two partners, three or more children with one partner and three or more children with two or more partners.

**Outcome assessment.**   Obesity is identified using Body Mass Index (BMI) derived from measured height and weight recorded by a health care professional at age 46, and calculated using the following formula: BMI = weight (kg) / height (m)$^2$. Obesity is defined as a BMI of 30 or over. Those with a BMI of 29 or below are combined into the reference category. Utilising BMI at age 46 allows for the temporal ordering of the variables to be established.

**Confounding variables.**   Confounders were considered based on a priori knowledge of factors in childhood that have been found to influence both the exposure (MPF) and the outcome (BMI). Confounders measured at birth include highest level of mother's or father's education (GCSE and below, or A level and above), father's occupational social class or mother's occupational social class if father was missing (skilled, partly-skilled, manual, managerial or professional), maternal age at birth (19 and under, 20–24, 25–29 and 30 or over), and a binary measure of maternal smoking during pregnancy.

Child health confounders measured at age 10 include diastolic blood pressure (DBP) and systolic blood pressure (SBP), measured by a health care professional and included as continuous variables; a binary variable assessing if the cohort member has any longstanding illnesses or disabilities; and the number of days the cohort member missed school for illness reported on a continuous scale. Child living standards is represented by: a binary variable assessing if the cohort member's home was affected by damp; a binary variable of overcrowding (person per room ratio above one); and a binary variable assessing if the parents of the cohort member received benefits. Child cognitive ability is assessed via a combined Friendly Math Test Score and Edinburgh Reading Test Score [50].

Adolescent confounders reported at age 16 include: a binary measure of parental separation (since birth); a binary measure assessing if the cohort member was in a romantic relationship at the time of the age 16 interview; and a binary measure of smoking status. Mental wellbeing is reported via the 24-question self-reported Malaise Inventory [47] (Cronbach's α = 0.79), an established scale measuring signs of psychological distress. Two measures of behaviour are the Locus of Control Scale–a self-reported 29-item questionnaire that measures internal and external locus of control (Cronbach's α = 0.66), and the Rutter Behavioural Scale, derived from observational behaviours of the cohort member (Cronbach's α = 0.80) [51]. Categorical ratings are divided into three levels of severity: "normal" scores less than the 80th percentile, "moderate" problem scores between the 80th and 95th percentile and "severe" problem scores above the 95th percentile. The final measure is a measure of self-esteem (LAWSEQ) [52] (Cronbach's α = 0.73).

## Adult mediators

Adult mediators at age 42 include smoking status (never, previous or current smoker) and the alcohol use disorders identification test (AUDIT)—an alcohol harm screening tool indicating possible alcohol dependency [53]. Mental wellbeing is reported via the shortened 8-question self-reported Malaise Inventory [51] (Cronbach's α = 0.79). Age at first birth categories for mothers include under 20, 20–29 and 30 and over; and for fathers under 24, 24–33 and 34 and over. Different categories for mothers and fathers are used due to the small number of fathers

who had a first birth as a teenager. Two measures of socioeconomic status include housing tenure (owned/mortgage, social rent or private rent/other), and highest academic qualification (No qualifications, GCSE or equivalent, and A levels and above).

## Statistical method

A directed acyclic graph (DAG) using DAGitty v3.0 is presented in S1 Fig in S1 File. The DAG guides a parsimonious approach towards the minimum sufficient set of variables in the models. As a result, variables including days of school missed due to poor health, SBP and DBP were excluded because they were an ancestor of the outcome but not related to the exposure. Having a girl/boyfriend at age 16, was an ancestor of the exposure but not related to the outcome and dropped from the models. Fig 1 is therefore drawn as a result of the DAG and shows the hypothesized association between the variables retained in the models.

We test the unadjusted association between number of biological children, MPF and the composite number of biological children-MPF variable and odds of obesity at midlife (Table 2). Secondly, odds ratios estimated from logistics regression examine if the number of biological children-MPF composite variable is associated with odds of obesity (Tables 3 and 4). Two children with one partner is selected as the baseline category. The parental, child and adolescent confounders are included and subsequently age at first birth and adult mediators are added into the model, creating a series of nested logistic regression models (Tables 3 and 4). Given theoretical predictions that the mechanisms underpinning MPF and health differ by gender, the analysis is conducted individually for mothers and fathers. Results are presented

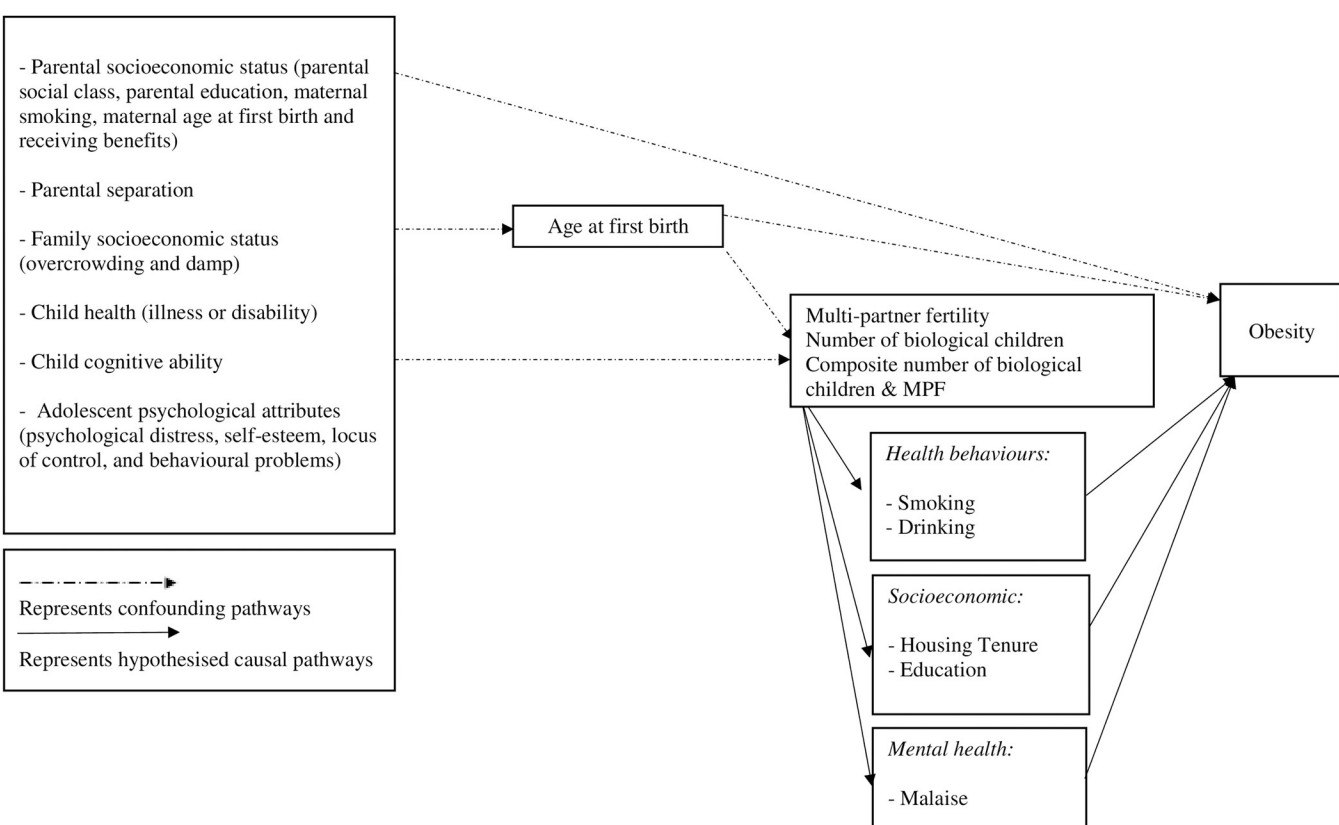

**Fig 1. Conceptual diagram showing the relationship between number of children, multi-partner fertility and a composite number of biological children-MPF variable and obesity risk.**

**Table 1. Sample characteristics by obesity status at age 46.** Fathers and mothers born in 1970.

| | | Fathers | | Mothers | |
|---|---|---|---|---|---|
| | | **% Obesity** | *Total (100%)* | **% Obesity** | *Total (100%)* |
| **MPF** | No | 34.5% | *2192* | 31.3% | *2513* |
| | Yes | 38.6% | *342* | 38.9% | *529* |
| **Number of biological children** | One child | 32.8% | *640* | 34.1% | *647* |
| | Two children | 35.3% | *1281* | 30.4% | *1593* |
| | Three or more children | 36.8% | *638* | 36.6% | *836* |
| **Number of biological children & MPF** | One child | 32.7% | *629* | 34.1% | *633* |
| | Two children & no MPF | 34.9% | *1157* | 29.4% | *1396* |
| | Three or more children & no MPF | 36.2% | *406* | 32.9% | *484* |
| | Two children & MPF | 40.7% | *113* | 38.5% | *187* |
| | Three or more children & MPF | 38.1% | *226* | 39.4% | *338* |
| *Parental confounders* | | | | | |
| Parental education | GCSE & below | 37.7% | *2643* | 35.4% | *2808* |
| | A level & above | 29.0% | *1130* | 26.5% | *1238* |
| Father's occupational social class | Unskilled | 38.3% | *180* | 42.8% | *201* |
| | Partly-skilled | 39.0% | *516* | 38.8% | *614* |
| | Manual | 36.9% | *2267* | 32.7% | *2378* |
| | Managerial | 28.8% | *565* | 28.8% | *631* |
| | Professional | 22.0% | *250* | 16.4% | *226* |
| Maternal age | 19 and under | 44.7% | *322* | 38.2% | *351* |
| | 20–24 | 35.5% | *1376* | 34.3% | *1460* |
| | 25–29 | 32.4% | *1202* | 31.0% | *1311* |
| | 30+ | 34.0% | *861* | 30.8% | *938* |
| Maternal smoking | Yes | 39.5% | *1604* | 36.6% | *1795* |
| | No | 31.9% | *2162* | 29.6% | *2261* |
| *Child confounders* | | | | | |
| Damp | Yes | 38.5% | *182* | 38.6% | *3435* |
| | No | 34.8% | *2051* | 32.2% | *184* |
| Receiving benefits | Yes | 40.4% | *716* | 35.6% | *859* |
| | No | 33.9% | *1952* | 32.2% | *2064* |
| Over-crowding | Yes | 38.6% | *1092* | 36.4% | *1163* |
| | No | 32.9% | *2449* | 31.2% | *2655* |
| Illness or disability | Yes | 37.9% | *934* | 35.1% | *870* |
| | No | 33.8% | *2318* | 31.7% | *2658* |
| *Adolescent confounders* | | | | | |
| Parental separation | Yes | 32.6% | *519* | 35.8% | *590* |
| | No | 34.7% | *1799* | 32.9% | *2102* |
| Smoking | Yes | 36.4% | *253* | 31.5% | *461* |
| | No | 33.4% | *1234* | 33.6% | *1745* |
| Behaviour | Normal | 33.3% | *1703* | 32.1% | *2012* |
| | Severe | 36.8% | *315* | 40.5% | *393* |
| Malaise Index | Low | 33.0% | *1200* | 32.3% | *1621* |
| | High | 32.6% | *129* | 35.1% | *353* |
| *Adult mediators* | | | | | |

*(Continued)*

**Table 1.** (Continued)

|  |  | Fathers | | Mothers | |
|---|---|---|---|---|---|
|  |  | % Obesity | *Total (100%)* | % Obesity | *Total (100%)* |
| Age at first birth | U24 | 40.2% | *333* |  |  |
|  | 24–33 | 36.7% | *1443* |  |  |
|  | 34+ | 30.2% | *688* |  |  |
|  | U20 |  |  | 44.0% | *248* |
|  | 20–29 |  |  | 36.2% | *1624* |
|  | 30+ |  |  | 25.5% | *1142* |
| Smoking status | Never | 36.0% | *1612* | 32.1% | *1826* |
|  | Previous | 36.0% | *982* | 33.9% | *1130* |
|  | Current | 32.0% | *860* | 33.1% | *849* |
| AUDIT–drinking | Yes | 33.2% | *1001* | 33.2% | *573* |
|  | No | 34.3% | *1850* | 31.6% | *2595* |
| Malaise | Low | 33.9% | *2581* | 30.5% | *2786* |
|  | High | 38.2% | *440* | 41.7% | *664* |
| Housing tenure | Owned/mortgage | 33.6% | *2649* | 30.1% | *2911* |
|  | Social rent | 38.0% | *287* | 49.4% | *470* |
|  | Private rent/other | 40.1% | *506* | 34.1% | *413* |
| Highest qualification | No qualifications | 38.0% | *1024* | 36.6% | *890* |
|  | GCSE | 38.8% | *1084* | 36.3% | *1253* |
|  | A level | 38.1% | *162* | 34.0% | *612* |
|  | Degree | 25.8% | *240* | 25.0% | *1056* |

The total sample size were based on the specific sample available at the sweep in which the variable was reported or measured.

**Table 2. Odds ratios from logistic regression models estimating the unadjusted associations between MPF, number of children and obesity odds for mothers and fathers.**

|  |  | Fathers (n-2940) | | | Mothers (n-3369) | | |
|---|---|---|---|---|---|---|---|
|  |  | Obesity | | | Obesity | | |
|  |  | OR | Sig. | 95% CI | OR | Sig. | 95% CI |
| **MPF** | No | REF | REF | REF | REF | REF | REF |
|  | Yes | 1.19 |  | (0.94, 1.51) | **1.35** | ** | **(1.12, 1.64)** |
| **Number of biological children** | One | 0.89 |  | (0.73, 1.10) | 1.18 |  | (0.97, 1.43) |
|  | Two | REF | REF | REF | REF | REF | REF |
|  | Three or more | 1.07 |  | (0.88, 1.30) | **1.26** | * | **(1.05, 1.50)** |
| **Number of biological children & MPF** | One child | 0.86 |  | (0.67, 1.12) | **1.24** | * | **(1.01, 1.51)** |
|  | Two children & no MPF | REF | REF | REF | REF | REF | REF |
|  | Three or more children & no MPF | 0.94 |  | (0.75, 1.19) | 1.19 |  | (0.95, 1.49) |
|  | Two children & MPF | 1.19 |  | (0.77, 1.82) | **1.45** | * | **(1.05, 1.99)** |
|  | Three or more children & MPF | 1.09 |  | (0.78, 1.53) | **1.51** | *** | **(1.18, 1.93)** |

\* P ≤ 0.05

\*\* P ≤ 0.01

\*\*\* P ≤ 0.001.

Base: No obesity.

Table 3. Odds ratios of the association between number of biological children-MPF composite variable and obesity odds for fathers.

| | | | Model 1 Unadjusted | | | Model 2 Parental confounders | | | Model 3 Child confounders | | | Model 4 Adolescent confounders | | | Model 5 Age at first birth | | | Model 6 Adult factors | | |
|---|---|---|---|---|---|---|---|---|---|---|---|---|---|---|---|---|---|---|---|---|
| | | | OR | Sig. | 95% CI | OR | Sig. | 95% CI | OR | Sig. | 95% CI | OR | Sig. | 95% CI | OR | Sig. | 95% CI | OR | Sig. | 95% CI |
| | Number of biological children & MPF | One child | 0.86 | | [0.67,1.12] | 0.81 | | [0.62,1.05] | 0.79 | | [0.61,1.03] | 0.82 | | [0.62,1.08] | 0.85 | | [0.65,1.13] | 0.86 | | [0.65,1.13] |
| | | Two children & No MPF | REF | REF | REF | REF | REF | REF | REF | REF | REF | REF | REF | REF | REF | REF | REF | REF | REF | REF |
| | | Three or more children & No MPF | 0.94 | | [0.75,1.19] | 0.91 | | [0.72,1.16] | 0.91 | | [0.72,1.16] | 0.96 | | [0.75,1.23] | 0.97 | | [0.76,1.24] | 0.96 | | [0.75,1.22] |
| | | Two children & MPF | 1.19 | | [0.77,1.82] | 1.02 | | [0.66,1.58] | 0.94 | | [0.60,1.46] | 1.06 | | [0.67,1.68] | 1.07 | | [0.67,1.70] | 1.10 | | [0.68,1.76] |
| | | Three or more children & MPF | 1.09 | | [0.78,1.53] | 0.93 | | [0.66,1.31] | 0.85 | | [0.60,1.21] | 0.87 | | [0.61,1.25] | 0.87 | | [0.60,1.26] | 0.87 | | [0.60,1.26] |
| Birth | Education | GCSE and below | | | | 1.31 | ** | [1.08,1.59] | 1.20 | | [0.98,1.47] | 1.18 | | [0.96,1.44] | 1.16 | | [0.95,1.43] | 1.11 | | [0.90,1.38] |
| | | A level and above | | | | REF | REF | REF | REF | REF | REF | REF | REF | REF | REF | REF | REF | REF | REF | REF |
| | Social Class | Unskilled | | | | 1.54 | | [0.92,2.57] | 1.31 | | [0.78,2.20] | 1.29 | | [0.76,2.20] | 1.28 | | [0.75,2.19] | 1.25 | | [0.73,2.16] |
| | | Partly Skilled | | | | 1.76 | ** | [1.15,2.69] | 1.55 | * | [1.01,2.38] | 1.55 | * | [1.00,2.41] | 1.54 | | [0.99,2.39] | 1.49 | | [0.95,2.33] |
| | | Manual | | | | 1.64 | * | [1.12,2.39] | 1.51 | * | [1.04,2.21] | 1.53 | * | [1.04,2.25] | 1.52 | * | [1.03,2.23] | 1.46 | | [0.99,2.16] |
| | | Managerial | | | | 1.33 | | [0.89,1.99] | 1.27 | | [0.85,1.91] | 1.31 | | [0.87,1.97] | 1.30 | | [0.86,1.95] | 1.27 | | [0.83,1.92] |
| | | Professional | | | | REF | REF | REF | REF | REF | REF | REF | REF | REF | REF | REF | REF | REF | REF | REF |
| | Maternal Age | 19 and under | | | | 1.42 | * | [1.06,1.91] | 1.37 | * | [1.01,1.85] | 1.40 | * | [1.02,1.92] | 1.39 | * | [1.02,1.91] | 1.36 | | [0.99,1.87] |
| | | 20–24 | | | | 1.03 | | [0.83,1.27] | 1.01 | | [0.81,1.24] | 1.01 | | [0.81,1.25] | 1 | | [0.80,1.25] | 0.98 | | [0.79,1.23] |
| | | 25–29 | | | | 0.99 | | [0.79,1.23] | 0.97 | | [0.78,1.21] | 0.97 | | [0.77,1.21] | 0.97 | | [0.77,1.21] | 0.95 | | [0.76,1.20] |
| | | 30+ | | | | REF | REF | REF | REF | REF | REF | REF | REF | REF | REF | REF | REF | REF | REF | REF |
| | Maternal Smoking | No | | | | REF | REF | REF | REF | REF | REF | REF | REF | REF | REF | REF | REF | REF | REF | REF |
| | | Yes | | | | 1.30 | ** | [1.11,1.52] | 1.25 | ** | [1.07,1.47] | 1.28 | ** | [1.08,1.50] | 1.27 | ** | [1.08,1.50] | 1.27 | ** | [1.08,1.50] |
| Age 10 | Cognitive ability | | | | | | | | 0.99 | ** | [0.99,1.00] | 0.99 | * | [0.99,1.00] | 0.99 | * | [0.99,1.00] | 1.00 | * | [0.99,1.00] |
| | Damp | No | | | | | | | REF | REF | REF | REF | REF | REF | REF | REF | REF | REF | REF | REF |
| | | Yes | | | | | | | 1.08 | | [0.76,1.52] | 1.16 | | [0.80,1.66] | 1.15 | | [0.80,1.66] | 1.14 | | [0.79,1.65] |
| | Illness or disability | No | | | | | | | REF | REF | REF | REF | REF | REF | REF | REF | REF | REF | REF | REF |
| | | Yes | | | | | | | 1.14 | | [0.94,1.37] | 1.14 | | [0.94,1.38] | 1.13 | | [0.93,1.37] | 1.14 | | [0.93,1.38] |
| | Overcrowding | No | | | | | | | REF | REF | REF | REF | REF | REF | REF | REF | REF | REF | REF | REF |
| | | Yes | | | | | | | 1.06 | | [0.89,1.27] | 1.06 | | [0.88,1.28] | 1.06 | | [0.88,1.28] | 1.07 | | [0.89,1.29] |
| | Benefits | No | | | | | | | REF | REF | REF | REF | REF | REF | REF | REF | REF | REF | REF | REF |
| | | Yes | | | | | | | 1.17 | | [0.95,1.45] | 1.24 | | [0.98,1.57] | 1.23 | | [0.97,1.56] | 1.26 | | [0.99,1.61] |

(Continued)

Table 3. (Continued)

| | | Model 1 Unadjusted | | | Model 2 Parental confounders | | | Model 3 Child confounders | | | Model 4 Adolescent confounders | | | Model 5 Age at first birth | | | Model 6 Adult factors | | |
|---|---|---|---|---|---|---|---|---|---|---|---|---|---|---|---|---|---|---|---|
| | | OR | Sig. | 95% CI | OR | Sig. | 95% CI | OR | Sig. | 95% CI | OR | Sig. | 95% CI | OR | Sig. | 95% CI | OR | Sig. | 95% CI |
| Age 16 | Self-Esteem | | | | | | | | | | 1.01 | | [0.99,1.02] | 1.01 | | [0.99,1.02] | 1.01 | | [0.99,1.02] |
| | Locus of control | | | | | | | | | | **0.95** | * | **[0.90,1.00]** | 0.95 | | [0.90,1.00] | 0.95 | | [0.90,1.00] |
| | Malaise Low | | | | | | | | | | REF | REF | REF | REF | REF | REF | REF | REF | REF |
| | Malaise High | | | | | | | | | | 0.63 | | [0.37,1.06] | 0.63 | | [0.37,1.07] | 0.64 | | [0.36,1.12] |
| | Rutter Behaviour Normal | | | | | | | | | | REF | REF | REF | REF | REF | REF | REF | REF | REF |
| | Rutter Behaviour Severe | | | | | | | | | | 0.94 | | [0.69,1.28] | 0.95 | | [0.70,1.29] | 0.98 | | [0.71,1.34] |
| | Parental Separation No | | | | | | | | | | REF | REF | REF | REF | REF | REF | REF | REF | REF |
| | Parental Separation Yes | | | | | | | | | | 0.78 | | [0.60,1.03] | 0.79 | | [0.60,1.04] | 0.78 | | [0.59,1.03] |
| | Smoker No | | | | | | | | | | REF | REF | REF | REF | EF | REF | REF | REF | REF |
| | Smoker Yes | | | | | | | | | | 1.05 | | [0.77,1.43] | 1.04 | | [0.76,1.42] | 1.25 | | [0.83,1.88] |
| Age 42 | Age at first birth 23 and under | | | | | | | | | | | | | 1.11 | | [0.81,1.53] | 1.1 | | [0.80,1.51] |
| | 24-33 | | | | | | | | | | | | | 1.17 | | [0.95,1.43] | 1.14 | | [0.92,1.40] |
| | 33+ | | | | | | | | | | | | | REF | REF | REF | REF | REF | REF |
| | Education No Education | | | | | | | | | | | | | | | | 1.22 | | [0.98,1.53] |
| | GCSE | | | | | | | | | | | | | | | | 1.32 | | [0.98,1.79] |
| | A Level | | | | | | | | | | | | | | | | 0.81 | | [0.61,1.09] |
| | Degree | | | | | | | | | | | | | | | | REF | REF | REF |
| | Smoking Never Smoked | | | | | | | | | | | | | | | | REF | REF | REF |
| | Used to smoke | | | | | | | | | | | | | | | | 0.9 | | [0.71,1.14] |
| | Smoker | | | | | | | | | | | | | | | | **0.6** | ** | **[0.43,0.82]** |
| | AUDIT Normal | | | | | | | | | | | | | | | | REF | REF | REF |
| | Problematic drinking | | | | | | | | | | | | | | | | 1.07 | | [0.87,1.31] |
| | Housing Tenure Own/mortgage | | | | | | | | | | | | | | | | REF | REF | REF |
| | Social rent | | | | | | | | | | | | | | | | 0.97 | | [0.69,1.35] |
| | Private rent & other | | | | | | | | | | | | | | | | 1.25 | | [0.96,1.63] |
| | Malaise Low | | | | | | | | | | | | | | | | REF | REF | REF |
| | Malaise High | | | | | | | | | | | | | | | | 1.09 | | [0.82,1.45] |
| | Observations | | | | | | | | | | | | | | | | | | 2940 |

* P ≤ 0.05
** P ≤ 0.01
*** P ≤ 0.001.
Base: No obesity.

Table 4. Odds ratios of the association between number of biological children-MPF composite variable and obesity odds for mothers.

| | | | Model 1 Unadjusted | | | Model 2 Parental confounders | | | Model 3 Child confounders | | | Model 4 Adolescent confounders | | | Model 5 Age at first birth | | | Model 6 Adult factors | | |
|---|---|---|---|---|---|---|---|---|---|---|---|---|---|---|---|---|---|---|---|---|
| | | | OR | Sig. | 95% CI | OR | Sig. | 95% CI | OR | Sig. | 95% CI | OR | Sig. | 95% CI | OR | Sig. | 95% CI | OR | Sig. | 95% CI |
| | Number of biological children & MPF | One child | 1.24 | * | [1.01,1.51] | 1.21 | | [0.99,1.48] | 1.20 | | [0.98,1.47] | 1.20 | | [0.98,1.48] | 1.30 | * | [1.05,1.60] | 1.28 | * | [1.03,1.58] |
| | | Two children & No MPF | REF | REF | | REF | REF | REF | REF | REF | REF | REF | REF | REF | REF | REF | REF | REF | REF | REF |
| | | Three or more children & No MPF | 1.19 | | [0.95,1.49] | 1.20 | | [0.95,1.50] | 1.16 | | [0.92,1.46] | 1.16 | | [0.92,1.46] | 1.08 | | [0.86,1.37] | 1.07 | | [0.85,1.36] |
| | | Two children & MPF | 1.45 | * | [1.05,1.99] | 1.31 | | [0.95,1.81] | 1.25 | | [0.90,1.73] | 1.25 | | [0.89,1.76] | 1.13 | | [0.80,1.59] | 1.1 | | [0.78,1.56] |
| | | Three or more children & MPF | 1.51 | *** | [1.18,1.93] | 1.35 | * | [1.05,1.74] | 1.28 | | [0.99,1.65] | 1.27 | | [0.98,1.66] | 1.12 | | [0.85,1.47] | 1.02 | | [0.77,1.35] |
| Birth | Education | GCSE and below | | | | 1.17 | | [0.97,1.40] | 1.08 | | [0.90,1.31] | 1.07 | | [0.89,1.30] | 1.05 | | [0.87,1.28] | 1.02 | | [0.84,1.24] |
| | | A level and above | | | | REF | REF | REF | REF | REF | REF | REF | REF | REF | REF | REF | REF | REF | REF | REF |
| | Social Class | Unskilled | | | | 3.16 | *** | [1.87,5.34] | 2.68 | *** | [1.57,4.57] | 2.61 | *** | [1.52,4.47] | 2.52 | *** | [1.47,4.32] | 2.46 | ** | [1.42,4.24] |
| | | Partly Skilled | | | | 2.75 | *** | [1.73,4.36] | 2.44 | *** | [1.53,3.89] | 2.39 | *** | [1.49,3.84] | 2.28 | *** | [1.42,3.65] | 2.10 | ** | [1.30,3.38] |
| | | Manual | | | | 2.24 | *** | [1.46,3.44] | 2.06 | ** | [1.33,3.17] | 2.00 | ** | [1.29,3.09] | 1.94 | ** | [1.25,3.00] | 1.87 | ** | [1.20,2.91] |
| | | Managerial | | | | 1.98 | ** | [1.26,3.10] | 1.92 | ** | [1.22,3.01] | 1.87 | ** | [1.19,2.95] | 1.85 | ** | [1.17,2.92] | 1.82 | * | [1.15,2.88] |
| | | Professional | | | | REF | REF | REF | REF | REF | REF | REF | REF | REF | REF | REF | REF | REF | REF | REF |
| | Maternal Age | 19 and under | | | | 1.20 | | [0.90,1.60] | 1.16 | | [0.87,1.54] | 1.18 | | [0.88,1.59] | 1.14 | | [0.84,1.53] | 1.13 | | [0.84,1.53] |
| | | 20-24 | | | | 1.09 | | [0.89,1.33] | 1.09 | | [0.89,1.33] | 1.13 | | [0.92,1.39] | 1.09 | | [0.89,1.34] | 1.1 | | [0.89,1.36] |
| | | 25-29 | | | | 1.03 | | [0.84,1.26] | 1.05 | | [0.85,1.29] | 1.08 | | [0.87,1.33] | 1.07 | | [0.87,1.32] | 1.08 | | [0.87,1.33] |
| | | 30+ | | | | REF | REF | REF | REF | REF | REF | REF | REF | REF | REF | REF | REF | REF | REF | REF |
| | Maternal Smoking | No | | | | REF | REF | REF | REF | REF | REF | REF | REF | REF | REF | REF | REF | REF | REF | REF |
| | | Yes | | | | 1.24 | ** | [1.07,1.44] | 1.20 | * | [1.03,1.39] | 1.23 | * | [1.05,1.43] | 1.22 | * | [1.04,1.42] | 1.20 | * | [1.03,1.41] |
| Age 10 | Cognitive ability | | | | | | | | 0.99 | *** | [0.99,1.00] | 0.99 | * | [0.99,1.00] | 1.00 | * | [0.99,1.00] | 1.00 | | [0.99,1.00] |
| | Damp | No | | | | | | | REF | REF | REF | REF | REF | REF | REF | REF | REF | REF | REF | REF |
| | | Yes | | | | | | | 1.25 | | [0.89,1.74] | 1.25 | | [0.89,1.74] | 1.22 | | [0.87,1.71] | 1.19 | | [0.84,1.68] |
| | Illness or disability | No | | | | | | | REF | REF | REF | REF | REF | REF | REF | REF | REF | REF | REF | REF |
| | | Yes | | | | | | | 1.1 | | [0.92,1.32] | 1.09 | | [0.91,1.32] | 1.09 | | [0.90,1.31] | 1.07 | | [0.89,1.30] |
| | Overcrowding | No | | | | | | | REF | REF | REF | REF | REF | REF | REF | REF | REF | REF | REF | REF |
| | | Yes | | | | | | | 1.07 | | [0.90,127] | 1.05 | | [0.89,1.25] | 1.04 | | [0.87,1.23] | 1.03 | | [0.86,1.23] |
| | Benefits | No | | | | | | | REF | REF | REF | REF | REF | REF | REF | REF | REF | REF | REF | REF |
| | | Yes | | | | | | | 1.01 | | [0.85,1.21] | 1.02 | | [0.84,1.25] | 1.00 | | [0.82,1.22] | 1.01 | | [0.82,1.24] |

*(Continued)*

**Table 4.** (Continued)

| | | | Model 1 Unadjusted | | | Model 2 Parental confounders | | | Model 3 Child confounders | | | Model 4 Adolescent confounders | | | Model 5 Age at first birth | | | Model 6 Adult factors | | |
|---|---|---|---|---|---|---|---|---|---|---|---|---|---|---|---|---|---|---|---|---|
| | | | OR | Sig. | 95% CI | OR | Sig. | 95% CI | OR | Sig. | 95% CI | OR | Sig. | 95% CI | OR | Sig. | 95% CI | OR | Sig. | 95% CI |
| Age 16 | Self-Esteem | | | | | | | | | | | 1.00 | | [0.99,1.01] | 1.00 | | [0.99,1.01] | 1.00 | | [0.99,1.01] |
| | Locus of Control | | | | | | | | | | | 0.97 | | [0.93,1.01] | 0.97 | | [0.94,1.01] | 0.98 | | [0.94,1.01] |
| | Malaise | Low | | | | | | | | | | REF | REF | REF | REF | REF | REF | REF | REF | REF |
| | | High | | | | | | | | | | 1.01 | | [0.75,1.37] | 1.01 | | [0.74,1.38] | 0.96 | | [0.70,1.32] |
| | Rutter Behaviour | Normal | | | | | | | | | | REF | REF | REF | REF | REF | REF | REF | REF | REF |
| | | Severe | | | | | | | | | | 1.36 | * | [1.03,1.79] | 1.34 | * | [1.02,1.77] | 1.29 | | [0.97,1.71] |
| | Parental Separation | No | | | | | | | | | | REF | REF | REF | REF | REF | REF | REF | REF | REF |
| | | Yes | | | | | | | | | | 0.93 | | [0.74,1.18] | 0.91 | | [0.72,1.16] | 0.90 | | [0.71,1.15] |
| | Smoker | No | | | | | | | | | | REF | REF | REF | REF | REF | REF | REF | REF | REF |
| | | Yes | | | | | | | | | | 0.69 | ** | [0.54,0.88] | 0.68 | ** | [0.53,0.86] | 0.65 | ** | [0.48,0.89] |
| Age 42 | Age at first birth | 23 and under | | | | | | | | | | | | | 1.67 | ** | [1.18,2.35] | 1.55 | * | [1.09,2.21] |
| | | 24–33 | | | | | | | | | | | | | 1.45 | *** | [1.20,1.74] | 1.38 | ** | [1.14,1.67] |
| | | 33+ | | | | | | | | | | | | | REF | REF | REF | REF | REF | REF |
| | Education | No Education | | | | | | | | | | | | | | | | 1.17 | | [0.94,1.44] |
| | | GCSE | | | | | | | | | | | | | | | | 1.21 | | [0.93,1.58] |
| | | A Level | | | | | | | | | | | | | | | | 0.91 | | [0.68,1.21] |
| | | Degree | | | | | | | | | | | | | | | | REF | REF | REF |
| | Smoking | Never Smoked | | | | | | | | | | | | | | | | REF | REF | REF |
| | | Used to smoke | | | | | | | | | | | | | | | | 1.17 | | [0.94,1.45] |
| | | Smoker | | | | | | | | | | | | | | | | 0.80 | | [0.61,1.07] |
| | AUDIT | Normal | | | | | | | | | | | | | | | | REF | REF | REF |
| | | Problematic drinking | | | | | | | | | | | | | | | | 1.08 | | [0.84,1.39] |
| | Housing Tenure | Own/ mortgage | | | | | | | | | | | | | | | | REF | REF | REF |
| | | Social rent | | | | | | | | | | | | | | | | 1.65 | *** | [1.28,2.12] |
| | | Private rent & other | | | | | | | | | | | | | | | | 1.07 | | [0.81,1.41] |
| | Malaise | Low | | | | | | | | | | | | | | | | REF | REF | REF |
| | | High | | | | | | | | | | | | | | | | 1.45 | *** | [1.17,1.81] |
| | | Observations | | | | | | | | | | | | | | | | | | 3369 |

* P ≤ 0.05
** P ≤ 0.01
*** P ≤ 0.001.
Base: No obesity.

after utilising Multiple Imputation (MI) by chained equations for missing observations at age 10, 16 and 42 [54]. 50 imputation cycles are constructed under the missing-at-random assumption [55–57], which has been found to be highly plausible in the British birth cohorts [58]. All variables are included in the imputation process. There was no missing data on the exposure variables, and the outcome is included in the imputed models, but imputed outcome values are not used. In S2 File we include the level of missing for each variable included in the final analytical sample for fathers and mothers separately.

## Results

### Descriptive results

Table 1 displays the proportion who have obesity at age 46 according to the different categories of our covariates. 14% of fathers experienced MPF prior to age 42 and 35% had obesity at age 46. 39% of fathers who experienced MPF had obesity at age 46, compared to 35% who had not experienced MPF. Fathers who had one child only had lower odds of obesity (33%). In comparison fathers who had two children with two different partners (39%) and fathers who had three or more children with two or more different partners (38%) had higher odds of obesity. 18% of mothers had experienced MPF prior to age 42 and 32% had obesity at age 46. 39% of mothers who experienced MPF had obesity at age 46, compared to 31% who had not experienced MPF. Mothers who had two children with the same partner (29%) had lower odds of obesity. In comparison, mothers who had two children with two different partners (39%) and mothers who had three or more children with two or more different partners (39%) had higher odds of obesity. Thus, the descriptive results demonstrate that parents who had more biological children and who had experienced MPF were more likely to have obesity at age 46.

Table 2 presents the unadjusted odds ratios (OR) of obesity according to gender and family dynamics (number of biological children, MPF and number of biological children-MPF) reported at age 42. For fathers, MPF, number of biological children and the composite number of biological children-MPF variable were not associated with obesity. Mothers who experienced MPF were 35% more likely to have obesity (95% CI 1.12–1.64) compared to mothers who had not experienced MPF. Mothers who had three or more children were 26% (95% CI 1.05–1.50) more likely to have obesity compared to mothers who had two children. When number of biological children is broken down by the experience of MPF, unadjusted estimates suggested that mothers who had three or more children but all with the same partner no longer had higher odds of obesity compared to mothers who had two children but all with the same partner. However, mothers who had two children with two partners and mothers who had three or more children with two or more partners were 45% (95% CI 1.05–1.99) and 51% (95% CI 1.18–1.93) more likely to have obesity compared to mothers who had two children with the same partner. Therefore, unadjusted estimates suggest that having a higher number of biological children with multiple partners, as opposed to a higher number of biological children regardless of the number of partners, is related to obesity odds for mothers.

### Multivariable regression analysis

Tables 3 and 4 present the OR of obesity at midlife according to the experience of the composite number of biological children-MPF variable from six separate models for fathers (Table 3) and mothers (Table 4). Each column, Model 1 to Model 6, represents a cumulative addition of a variable or group of variables into the nested regression models. For fathers, in unadjusted (Model 1) and adjusted models (Model 2–6) (Table 3), there was no significant association between number of biological children-MPF and the odds of obesity, although the findings regarding other risk factors of obesity are consistent with previous research.

Table 4 demonstrates that the significant unadjusted association (Model 1) between mothers with one child and mothers who had two children with two different partners and higher odds of obesity were attenuated when parental confounders (parental education, parental social class, maternal age at birth and maternal smoking) were included (one child–OR 1.21 95% CI 0.99–1.48; two children with two partners–OR 1.31 95% CI 0.95–1.81) (Model 2). The increased odds of obesity in mothers who had three or more children with two or more partners was attenuated when confounders measured at age 10 (cognitive ability, damp, illness or disability, overcrowding and parent's receiving benefits) were included (OR 1.28 95% CI 0.99–1.65) (Model 3). In model 5, the inclusion of age at first birth resulted in a significant association between mothers who had only one child and obesity (OR 1.30 95% CI 1.05–1.60). This suggest that age at first birth may be acting as a suppressor for this group, and this may be because on average the older the age at first birth the lower the rates of obesity. However, mothers with one child only are also likely to have an older age at first birth. So, when age at first birth is controlled for, the higher odds of obesity is 'revealed'.

## Discussion

High parity, early age at first birth and partnership separation have previously been associated with higher risk of obesity [8–13]. However, the implication of complex family dynamics on the odds of obesity at midlife has yet to be considered. This study analysed whether midlife obesity was associated with the experience of childbearing across partnerships, incorporating the number of biological children and MPF, for fathers and mothers born in the UK in 1970.

For fathers, there was no association between number of biological children, MPF and obesity. Mothers with one child, mothers with two children with two partners, and mothers with three or more children with two or more partners had higher odds of obesity compared to mothers who had two children both with the same partner. In adjusted analyses mothers with one child remained at higher odds of obesity, compared to mothers who had two children both with the same partner. However, all other associations were attenuated when parental background factors and childhood characteristics were accounted for. It is also likely that other unmeasured factors may contribute to the relationship between mothers who have one child and increased odds of obesity, these include pre-pregnancy health status, unfavourable prepregnant lipid profiles [59], some hormonal contraception [60], the potential protective effect of future pregnancies [61], and pre-pregnancy obesity or elevated blood pressure [62].

We hypothesised that MPF would be associated with midlife obesity, given that it often involves an early age at first birth, high parity and multiple partnership dissolution—three factors that have been found to be independently associated to BMI as part of cardiovascular risk [8,9,11,26,63–65]. However, our study provides mixed support for this hypothesis. With regards to the number of children, results support Sironi et al., [66] who found a weak association between parity and biomarkers of health (fibrinogen, C-reactive protein, glycated haemoglobin, cholesterol ratio, high blood pressure, obesity and metabolic syndrome) at midlife. For mothers, the findings in relation to one child and obesity supports a body of research that has demonstrated a relationship between parity one and poorer health outcomes including coronary heart disease, mortality and diabetes [8,67–72]. This relationship may reflect ill-health among mothers of parity one including pregnancy complications that may result in infertility [8,69]. Additionally, it is important to consider the complicated relationship between gynaecological factors and obesity. Poor nutrition and weight gain are not only likely to increase the risk of gynaecological conditions that may affect fertility [73], but gynaecological conditions may also influence further weight gain [74,75].

Our results contradict previous research that have found high parity to be associated to health outcomes including obesity [8–10]. One explanation is that previous research tends to find an association to those who are of parity four or higher. However, due to the small number of people who were of parity four or over and had experienced MPF, we could only consider those who were of parity three or higher. A further explanation is that with the exception of Sironi et al., [66] the majority of previous research has not been able to account for the same level of confounding as was done here. Conversely, this study goes beyond previous research that has focused exclusively on the role of parity to demonstrate that it might not be higher numbers of biological children that is related to the odds of obesity; but rather the fact that those with higher numbers of biological children are additionally more likely to experience MPF and this is highlighted in Table 2.

Although this paper did not provide any direct evidence about the pathways that linked the composite number of biological children-MPF variable to obesity, most of the observed relationships were attenuated by a combination of early life course confounders relating to parental social class, maternal smoking, child cognitive ability and psychological characteristics. Additionally, if researchers consider that, after controlling for number of children, the consequences of MPF are predominantly linked to stress that may arise from changes to the social and economic environment (e.g., partnership formation and dissolution, the family environment, parenting arrangements and decline in social and economic resources). Then, the findings that mothers who had two children with two different partners, or three or more children with two or more partners were at higher odds of obesity compared to mothers who had three or more children with one partner, provides some support for a pathway where stress related to MPF may have a biological effect on obesity. This supports research suggesting social or behavioural factors explain the relationship between family dynamics different to the ones we considered and health at midlife [76].

Research has demonstrated that parental background factors including social class, education and partnership dissolution, and childhood experiences such as physical health, cognition and psychological attributes are associated with both family dynamics [77–80] and poorer health [81–87]. We considered parental, child and adolescent characteristics before partnership formation or childbearing (aged 16 and under) that could have attenuated the associations we considered in this paper. We adjusted for parental and family socioeconomic status given socioeconomic status has consistently been found to predict demographic [77] and health outcomes [82,83,88]. Parental separation was considered because it is associated with early childbearing and partnership dissolution [89], and has been found to increase the risk of BMI [90,91]. We additionally considered the cohort members childhood psychological attributes as they are a confounder of both demographic and physical health outcomes [81,84–86]. Psychological attributes are linked to mental health [65], and poor mental health has been found to increase the risk of physical ill-health [66] and is associated to early parenthood [67]. Finally, child cognitive ability was considered as it is related to future health outcomes [68–70] and has been found to increase the risk of separation, early initiation of sexual activity and early childbearing [49].

Overall, our study found support for selection suggesting that parental background factors and childhood conditions influence both MPF and obesity. This finding is consistent with Perelli-Harris and Styrc [92] who found that amongst the BCS70, childhood selection attenuated differences in cohabitation and marriage on mental wellbeing. The results demonstrate that those with greater childhood resources may continue to lead stable family lives, whereas those with fewer childhood resources may experience greater family instability, complexity and poorer health. We provide evidence to suggest that factors in early childhood may precondition individuals towards complex family trajectories in adulthood. Moreover, McLanahan's

[93] 'diverging destinies' thesis argues that complex, unstable and 'non-traditional' family dynamics are associated with poorer outcomes. However, the results presented here provide little support for this hypothesis once selection is accounted for. Using prospective longitudinal data, we demonstrated that selection mechanisms that date back to childhood, and occurred before fertility or partnership trajectories, confound the differential effects of number of biological children and MPF; providing further evidence on how early childhood conditions are important for determining future outcomes [94–97].

This study demonstrated differences for mothers and fathers, and these differences could result from three explanations. Firstly, biological pathways associated with pregnancy and childbirth are not relevant for fathers. There are also important biological reproductive confounders potentially influencing obesity at midlife beyond pregnancy characteristics. For example early age at menarche and the menopause transition may be associated to increased obesity risk at midlife [38,39], whilst breastfeeding may have some long-term protective effect on the risk of excessive weight and abdominal obesity [40].

Secondly, mothers will likely have greater responsibility for children, especially following a partnership dissolution and in the context of MPF mothers may have more responsibility managing blended families. It could therefore be hypothesised that mothers may experience greater levels of stress and are more likely to be exposed to a decline in economic and social resources. Finally, the effect of health selection may be greater for mother's than father's given that a mother's health is likely to play a larger role in fertility decisions than a father's health [98].

## Strengths and limitations

The BCS70 is a large British cohort study that despite attrition has retained a large sample with detailed, prospectively collected information about parental background, childhood and adolescent circumstances which can be linked to outcomes in adulthood. The BCS70, unlike many surveys, collected detailed information on fertility and partnership experiences of men as well as women, affording the opportunity to explore gender differences and utilises measured obesity. However, estimates of MPF for men are likely to be less accurate than for women: men are particularly likely to underestimate their fertility, especially amongst those that are disadvantaged [99]; and whilst age 42 may be towards the end of childbearing for most women, men can continue their childbearing into midlife and beyond. Further, disruptive events across the life course, such as partnership dissolution and divorce, have been found to predict loss of follow-up [100], and therefore we are likely to have presented conservative estimates of MPF. A further issue with using observational studies as was done here, is that causality cannot be established.

The multiple imputation method assumed that variables at birth and age 10 generally predicted characteristics at age 16 (the sweep with the highest level of attrition); this approach left little development throughout adolescence and may have resulted in the overestimation of the effect of the early life factors. Additionally, the BCS70 cohort is representative of a cohort of children born in 1970, and as such does not reflect the ethnically diverse population of the UK today. Further, despite using measured rather than self-reported BMI there remains limitations including overestimating body fat in those who have muscular builds. There are also biological and pharmacological factors that may play a role in weight gain that we are unable to consider, these factors include oral contraception, hormonal therapy use, menopause, and the uses of certain medication (such as lipid lowering drugs, thyroid therapy, anxiolytics) [39,101,102].

Additionally, this paper did not consider BMI in childhood despite childhood BMI being highly correlated with BMI in adulthood [103]. This was because BMI cannot be considered a

confounder given the pathways between childhood BMI and MPF are speculative at best. Instead, childhood BMI should perhaps be considered an effect modifier for adult BMI. In S3 and S4 Files we compare the nested regression models testing the relationship between MPF and obesity for mothers and fathers including and excluding child BMI measured at age 10. However, it should be noted that the cut-off points for obesity differ between children and adults [104]. As shown, the addition of child BMI did not alter the results. There is also a genetic component to the odds of obesity, this study could not address. Around 40–70% of inter-individual variability in BMI has been attributed to genetic factors [105–107] and research suggests that genetically predisposed people are at greater risk of higher BMI [107,108]. However, researchers should be wary of overly-simplistic genetic interpretations given that genetic factors interact with environmental factors. Additionally, there was no accurate data on either the diet of the cohort members or their levels of physical exercise at age 42.

Other factors that require further research include personality type and health behaviour. Research has indicated that individuals who experience disruptive family life courses tend to have more risk taking and impulsive personalities [109,110]. Research has also found an association between personality types, risky health behaviour and poorer health outcomes [111,112]. Finally, we were not able to consider the child custody arrangements for parents who experience MPF. We did not differentiate between children living with the mother or father post separation; nor did we consider the timing of when separation, re-partnering and childbearing occurred. Finally, this study did not consider how a child's ability to cope with disruptive family events will likely impact the health of the parents.

## Conclusion

On the evidence of this study, number of biological children and MPF are not associated to the odds of obesity after accounting for confounding by lifecourse factors. Any relationship between parity and MPF with later obesity is likely to operate through selection. Research should continue to explore and understand the role of early childhood conditions in shaping future demographic and health outcomes. We did demonstrate that mothers who have one child only may be at increased odds of obesity, however, this relationship could be due to a number of unmeasured contributing factors and because age at first birth was acting as a suppressor for this group. This study has provided evidence supporting how early childhood conditions are important for determining future demographic and health outcomes supporting other evidence to focus on lifecourse determinants of health in policy considerations.

## Supporting information

**S1 File. Supplementary materials 1.** A directed acyclic graph illustrating the relationship between MPF at age 42 and obesity at age 46.
(DOCX)

**S2 File. Supplementary materials 2.** Level of missing for each variable in the final analytical sample, for fathers and mothers.
(DOCX)

**S3 File. Supplementary materials 3.** Odds ratios of obesity according to whether or not the cohort member had experienced multi-partner fertility, where childhood BMI is included in analysis. Baseline outcome: No multi-partner fertility.
(DOCX)

**S4 File. Supplementary materials 4.** Odds ratios of obesity according to whether or not the cohort member had experienced multi-partner fertility and excluding childhood BMI. Baseline

outcome: No multi-partner fertility.
(DOCX)

## Acknowledgments

For the purpose of open access, the author has applied a CC BY public copyright licence to any Author Accepted Manuscript version arising from this submission.

## Author Contributions

**Conceptualization:** Sebastian Stannard, Ann Berrington, Nisreen A. Alwan.

**Data curation:** Sebastian Stannard.

**Formal analysis:** Sebastian Stannard.

**Investigation:** Sebastian Stannard.

**Methodology:** Sebastian Stannard.

**Project administration:** Sebastian Stannard.

**Resources:** Sebastian Stannard.

**Software:** Sebastian Stannard.

**Supervision:** Ann Berrington, Nisreen A. Alwan.

**Validation:** Sebastian Stannard.

**Visualization:** Sebastian Stannard.

**Writing – original draft:** Sebastian Stannard, Ann Berrington, Nisreen A. Alwan.

**Writing – review & editing:** Sebastian Stannard, Ann Berrington, Nisreen A. Alwan.

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
