## [Decision Letter · Decision Letter 0]

11 Jan 2023

PONE-D-22-29433Exploring the associations between number of children, multi-partner fertility and risk of obesity at midlife: Findings from the 1970 British Cohort Study (BCS70)PLOS ONE

Dear Dr. Stannard,

Thank you for submitting your manuscript to PLOS ONE. After careful consideration, we feel that it has merit but does not fully meet PLOS ONE’s publication criteria as it currently stands. Therefore, we invite you to submit a revised version of the manuscript that addresses the points raised during the review process.

We look forward to receiving your revised manuscript.

Kind regards,

Emily W. Harville

Academic Editor

PLOS ONE

Journal Requirements:

2. Please provide additional details regarding participant consent. In the ethics statement in the Methods and online submission information, please ensure that you have specified what type you obtained (for instance, written or verbal, and if verbal, how it was documented and witnessed). If your study included minors, state whether you obtained consent from parents or guardians. If the need for consent was waived by the ethics committee, please include this information

3. We note that you have referenced (ie. Bewick et al. [5]) which has currently not yet been accepted for publication. Please remove this from your References and amend this to state in the body of your manuscript: (ie “Bewick et al. [Unpublished]”) as detailed online in our guide for authors

"This work was partly funded by the Economic and Social Research Council (UK) [ES/P000673/1], and supported by ESRC Centre for Population Change (UK). For the purpose of open access, the author has applied a CC BY public copyright licence to any Author Accepted Manuscript version arising from this submission."

Additional Editor Comments (if provided):

Please address or refute the reviewers' comments.

Reviewers' comments:

Reviewer's Responses to Questions

**Comments to the Author**

1. Is the manuscript technically sound, and do the data support the conclusions?

Reviewer #1: Partly

Reviewer #2: Yes

2. Has the statistical analysis been performed appropriately and rigorously? 

Reviewer #1: Yes

Reviewer #2: Yes

3. Have the authors made all data underlying the findings in their manuscript fully available?

Reviewer #1: Yes

Reviewer #2: Yes

4. Is the manuscript presented in an intelligible fashion and written in standard English?

Reviewer #1: Yes

Reviewer #2: Yes

5. Review Comments to the Author

Reviewer #1: General comments

The present manuscript explores the associations between number of children, multi-partner fertility and risk of obesity at midlife: Findings from the 1970 British Cohort Study (BCS70). While focusing mainly on sociological and -related factors, study tries to explain the complex interplay of multiple factors that could play a role in obesity risk.

At present, the content of the different sections is not sufficient. Indeed, the Discussion is insufficiently documented, lacking important biological factors, which should be clearly acknowledged in the Limitation section. The section devoted to the materials and methods is currently not sufficiently described and doesn’t allow the reproducibility of the analyses (e.g. lack of details on BMI calculations and exact number of missing data by variables.). Methodologically, measuring risk of obesity in women at the age of 46 (one-time measuring point) basically at the menopausal transition is problematic in itself, given the effect of menopause on obesity in women.

Specific comments

Abs & Introduction

Line 27: the following sentence is false: “the relationship may not be causal”. This is an observational study, and therefore all the observed associations are non-causal. This should be also clearly stated in Limitation section. As for the line 27, questionable part should be replaced by: “suggesting that the part of the association is explained by confounders” or similar interpretation.

Methods & Results

Lines112-113: What about miscarriages and/or abortions, how are those handled? This should be clearly stated.

Lines 125-127: Further, in order to provide appropriate reproducibility of the study, please provide information on the exact formula used for BMI calculation.

Line 131: Please, provide information how are the used confounders established: a priori (based on what?), or after DAGs?

Lines186-188:

It is not currently clear from the stated what was the exact number of missing cases by used variables prior to MI method was applied. Please, provide exact N (%) for the missing data. Please, explain how was the “missing at random” assumption obtained?

Discussion

Lines 397-.398 The authors state: “We did demonstrate that mothers who have one child only may be at increased odds of obesity, however this association is likely to be partly because age at first birth is a suppressor for this group.

The study conclusion fails to comment and provide other important contributing factors present in the literature (for example: Pirnat A, DeRoo LA, Skjærven R, et al

Women’s prepregnancy lipid levels and number of children: a Norwegian prospective population-based cohort study BMJ Open 2018;8:e021188. doi: 10.1136/bmjopen-2017-021188

The Discussion needs more clear acknowledgment of biological factors that play significant role in development of obesity.

Are there available data on, duration of oral contraception/hormonal therapy use, sex hormone status, menopause, used medicines (lipid lowering drugs, thyroid therapy, anxiolytics) – all of which affects BMI? If not, this should be clearly stated in the Discussion section, under limitations.

Reviewer #2: Review

Generally, the manuscript was well written and the results were clearly presented.

1. Introduction was understandable, the individual paragraphs followed each other logically and clearly defined the importance and the need of the presented study.

2. An array of confounders which predispose individuals to obesity is very comprehensive.

3. A very extensive study sample ensures the accuracy of the results.

4. Statistical methods are suited to research questions posed at the study.

5. Results of the study are well presented and clearly explained.

6. Research question is clearly defined and appropriately answered.

Despite the above facts, I have a few following minor comments for the authors:

1.Various adult mediators at age 42 have been selected with the assumption that these affect obesity at age of 46. However, it is not clear from the article why the authors decided on the age of 42. Is there any scientific explanation for this selected age?

2.The authors revealed an important gender differences in the effect of parity as well as MPF on obesity manifestation at midlife. They correctly explained the reasons for this fact in the last paragraph of the section Discussion. However, some additional important data about female reproductive factors should be also considered and discussed. In the context with the recent findings that obesity at midlife is significantly associated not only with pregnancy characteristics, but also with other female reproductive factors, such as age at menarche and breastfeeding (Vorobelova et al. 2022, Ciesla et al. 2021). And thus, they present an important confounders influencing obesity at midlife.

3.In section „Strengths and limitations“ is written (lines 374-376): „Supplementary Materials S2 and S3 we compare the nested regression models testing the relationship between MPF and obesity for mothers and fathers including and excluding child BMI measured at age 10.“

However, neither in the methods nor in any other paragraph was there any information about the body mass index cutoff points for evaluation of nutritional status in 10 years old children. It is known that the cut off points differ between children and adults as established Cole et al. (2000). For example, the cut off point for BMI for obesity in 10-years old girls is 24,11 kg/m2. Therefore, I recommend to add the reference into the text.

(Establishing a standard definition for child overweight and obesity worldwide: international survey, BMJ 2000; 320) doi: https://doi.org/10.1136/bmj.320.7244.1240

4.The year of publication is missing in the reference with the number 99 (Light A, Ahn T. Divorce as risky behavior. Demography. 47, 895–921).

6. PLOS authors have the option to publish the peer review history of their article (what does this mean?). If published, this will include your full peer review and any attached files.

Reviewer #1: No

Reviewer #2: No

---

## [Author Response · Author response to Decision Letter 0]

9 Feb 2023

Reviewer 1

1. The present manuscript explores the associations between number of children, multi-partner fertility and risk of obesity at midlife: Findings from the 1970 British Cohort Study (BCS70). While focusing mainly on sociological and -related factors, study tries to explain the complex interplay of multiple factors that could play a role in obesity risk. At present, the content of the different sections is not sufficient. Indeed, the Discussion is insufficiently documented, lacking important biological factors, which should be clearly acknowledged in the Limitation section. The section devoted to the materials and methods is currently not sufficiently described and doesn’t allow the reproducibility of the analyses (e.g. lack of details on BMI calculations and exact number of missing data by variables.). Methodologically, measuring risk of obesity in women at the age of 46 (one-time measuring point) basically at the menopausal transition is problematic in itself, given the effect of menopause on obesity in women.

Thank you for this detailed comment, the issues raised have been addressed in the reviewers comments below. We have addressed the issue of lacking biological factors in comment 7, and we have improved the reproducibility of the study in comments 4, 5 and 6. We thank you for bringing to our attention the issue of the menopause. We agree that the menopause transition may increase the risk of obesity for all women (so we are perhaps overestimating obesity prevalence in women). However, we do not have a theoretical reason why the menopause would influence our analysis. There is no clear reason why menopause would affect MPF women differently, especially given a number of papers have found no relationship between parity and age of menopause (Sun et al., 2021; Rizvanovic et al., 2013). 

2. Line 27: the following sentence is false: “the relationship may not be causal”. This is an observational study, and therefore all the observed associations are non-causal. This should be also clearly stated in Limitation section. As for the line 27, questionable part should be replaced by: “suggesting that the part of the association is explained by confounders” or similar interpretation.

Thank you, we agree that a limitation of observational studies is that we cannot infer causal associations, and we have changed the wording on line 27 to make this clearer. As suggested, we now state that part of the association is explained by confounders rather than previously states that ‘the relationship may not be causal’. We have also stated this limitation on lines 376-377. 

3. Lines 112-113: What about miscarriages and/or abortions, how are those handled? This should be clearly stated.

Thank you, our MPF estimate only considers live births and therefore excludes miscarriages and abortions. We have now made this clearer on lines 113-114. It should also be noted that the fertility history of the cohort members including miscarriages, stillbirths and abortions are insufficiently detailed to incorporate these measures into our estimates of MPF (given we would require accurate reporting of the dates of these fertility events). It would be interesting if future research built in the role of pregnancies that did not result in a live birth, however this is heavily reliant on data availability and quality. 

4. Lines 125-127: Further, in order to provide appropriate reproducibility of the study, please provide information on the exact formula used for BMI calculation.

Thank you, we have now included this formula on line 129. 

5. Line 131: Please, provide information how are the used confounders established: a priori (based on what?), or after DAGs?

The confounders were considered based on a priori knowledge of factors in childhood that have been found to influence both the exposure (MPF) and the outcome (BMI) by published evidence. We have added additional information highlighting this on lines 134-135. 

6. Lines186-188: It is not currently clear from the stated what was the exact number of missing cases by used variables prior to MI method was applied. Please, provide exact N (%) for the missing data. Please, explain how was the “missing at random” assumption obtained?

Thank you, we have included an additional table in Supplementary Materials 3, this includes the exact number of missing cases by used variable in the final analytical sample for men and women separately. 

Thank you for your comment on missing at random. We agree that in order to conduct multiple imputation, data must be missing at random – that is the underlying reasons for data being missing are related to know factors (Moniek et al., 2013; Stern et al., 2009). Missing at random therefore allows prediction of the missing values based on the participants with complete data (Stern et al., 2009). UCL and the Centre for Longitudinal Studies, who are the data curators for the British birth cohort studies have spent considerable time and effort reporting on how to handle missing data in the BCS70. Their research supports our decision to conduct multiple imputation under the missing at random assumption given that missing values and the observed values can be explained by the rich information available in the cohort, for more information please see CLS | Handling missing data (ucl.ac.uk). 

7. Lines 397-.398 The authors state: “We did demonstrate that mothers who have one child only may be at increased odds of obesity, however this association is likely to be partly because age at first birth is a suppressor for this group.

The study conclusion fails to comment and provide other important contributing factors present in the literature (for example: Pirnat A, DeRoo LA, Skjærven R, et al Women’s prepregnancy lipid levels and number of children: a Norwegian prospective population-based cohort study BMJ Open 2018;8:e021188. doi: 10.1136/bmjopen-2017-021188

Thank you, we have re-written the concluding lines (419-421) and we now allude to the fact that the relationship between one child and obesity may be due to a number of additional contributing factors. We have also added additional information in the discussions (lines 274-278) and provide examples of other contributing factors to this relationship. 

8. The Discussion needs more clear acknowledgment of biological factors that play significant role in development of obesity. Are there available data on, duration of oral contraception/hormonal therapy use, sex hormone status, menopause, used medicines (lipid lowering drugs, thyroid therapy, anxiolytics) – all of which affects BMI? If not, this should be clearly stated in the Discussion section, under limitations.

Thank you for bringing this to our attention. Unfortunately, the design of the BCS70 datasets means we do not have the detailed biological factors highlighted by the reviewer, however we have now stated this limitation on pages 384-391. 

Reviewer 2: 

1. Various adult mediators at age 42 have been selected with the assumption that these affect obesity at age of 46. However, it is not clear from the article why the authors decided on the age of 42. Is there any scientific explanation for this selected age? 

We selected age 42 for a number of reasons. Firstly, we wanted to ensure that the temporal ordering of the variables could be established and given that we utilised BMI at age 46 the mediators needed to be recorded prior to this sweep of data collection. We are aware that a limitation of our approach is that although the mediators were measured temporally after MPF had occurred, we cannot rule out the possibility that some of these mediators were present prior to MPF. Secondly, MPF is the result of several interrelated life events – to experience MPF a person must have experience two or more live births. Secondly, they are likely to have experienced separation from the biological parent of one or more of their children and may have lived as a lone parent. Thirdly, MPF generally involves re-partnering given that most people who experience MPF will experience this across multiple coresidential partnerships (although it is possible to experience MPF outside of any coresidential partnerships). To accumulate all these life event will take considerable time, we therefore required the mediators to be recorded later in the life course to allow for these life events (associated to MPF) to accrue and for the impact of these life events on the mediators to become apparent. Finally, we chose age 42 given this may be the end of childbearing for most women and therefore estimates of MPF (for women) were likely to remain stable over time. However, if we had chosen an earlier timepoint (age 34 or 38) women were still likely to be having children. 

2. The authors revealed an important gender differences in the effect of parity as well as MPF on obesity manifestation at midlife. They correctly explained the reasons for this fact in the last paragraph of the section Discussion. However, some additional important data about female reproductive factors should be also considered and discussed. In the context with the recent findings that obesity at midlife is significantly associated not only with pregnancy characteristics, but also with other female reproductive factors, such as age at menarche and breastfeeding (Vorobelova et al. 2022, Ciesla et al. 2021). And thus, they present an important confounders influencing obesity at midlife.

Thank you for this important comment, we have now expanded this section of the discussions (lines 352-356) and now discuss other important female reproductive factors (aside from pregnancy characteristics) that are important confounders of obesity. We have also included some of these factors where we discuss gender differences in the discussions (line 78-79).

3. In section „Strengths and limitations“ is written (lines 374-376): „Supplementary Materials S2 and S3 we compare the nested regression models testing the relationship between MPF and obesity for mothers and fathers including and excluding child BMI measured at age 10.“ However, neither in the methods nor in any other paragraph was there any information about the body mass index cut-off points for evaluation of nutritional status in 10 years old children. It is known that the cut off points differ between children and adults as established Cole et al. (2000). For example, the cut off point for BMI for obesity in 10-years old girls is 24,11 kg/m2. Therefore, I recommend to add the reference into the text. (Establishing a standard definition for child overweight and obesity worldwide: international survey, BMJ 2000; 320) 

Thank you for bringing this to our attention it is an important point. We now included the reference stated and mention that cut-off points for BMI in childhood differ from those used in adulthood (line 394). However, it should be noted there still remains some debate about how best to define obesity status in childhood (Brown et al., 2017; Serra-Majem et al., 2007; Sweeting, 2007; Neovius et al., 2004), and that the cut-off for defining obesity in childhood remains disputed (Serra-Majem et al., 2007; Sweeting, 2007; Neovius et al., 2004). Therefore, given the aim of including child BMI in the supplementary materials was to understand it’s confounding role, we decided to include child BMI as a continuous measure. 

4. The year of publication is missing in the reference with the number 99 (Light A, Ahn T. Divorce as risky behaviour. Demography. 47, 895–921).

Thank you we have now included this missing year of publication. 

References: 

Elise C. Brown, J. Lon Kilgore, Duncan S. Buchan & Julien S. Baker (2017). A criterion-referenced assessment is needed for measuring child obesity, Research in Sports Medicine, 25:1, 108-110

Serra-Majem, L., Ribas-Barba, L., Pérez-Rodrigo, C., Ngo, J., & Aranceta, J. (2007). Methodological limitations in measuring childhood and adolescent obesity and overweight in epidemiological studies: Does overweight fare better than obesity? Public Health Nutrition, 10(10A), 1112-1120.

Sweeting, H.N. (2007). Measurement and Definitions of Obesity In Childhood and Adolescence: A field guide for the uninitiated. Nutr J 6.

Neovius, M., Linné, Y., Barkeling, B. and Rössner, S. (2004), Discrepancies between classification systems of childhood obesity. Obesity Reviews. 5: 105-114.

Moniek C.M. de Goeij, Merel van Diepen, Kitty J. Jager, Giovanni Tripepi, Carmine Zoccali, Friedo W. Dekker. (2013). Multiple imputation: dealing with missing data, Nephrology Dialysis Transplantation, 28(10): 2415–2420.

Sterne JAC, White IR, Carlin JB, Spratt M, Royston P, Kenward MG, Wood AM, Carpenter JR: Multiple imputation for missing data in epidemiological and clinical research: potential and pitfalls. BMJ. 2009;338:157–60.

Sun X, Zhang R, Wang L, et al. Association Between Parity and the Age at Menopause and Menopausal Syndrome in Northwest China. Asia Pac J Public Health. 2021;33(1):60-66.

Rizvanovic M, Balic D, Begic Z, Babovic A, Bogadanovic G, Kameric L. Parity and menarche as risk factors of time of menopause occurrence. Med Arch. 2013;67(5):336-338.

---

## [Decision Letter · Decision Letter 1]

23 Feb 2023

Exploring the associations between number of children, multi-partner fertility and risk of obesity at midlife: Findings from the 1970 British Cohort Study (BCS70)

PONE-D-22-29433R1

Dear Dr. Stannard,

We’re pleased to inform you that your manuscript has been judged scientifically suitable for publication and will be formally accepted for publication once it meets all outstanding technical requirements.

Kind regards,

Emily W. Harville

Academic Editor

PLOS ONE

Additional Editor Comments (optional):

Reviewers' comments:

Reviewer's Responses to Questions

**Comments to the Author**

1. If the authors have adequately addressed your comments raised in a previous round of review and you feel that this manuscript is now acceptable for publication, you may indicate that here to bypass the “Comments to the Author” section, enter your conflict of interest statement in the “Confidential to Editor” section, and submit your "Accept" recommendation.

Reviewer #1: All comments have been addressed

Reviewer #2: All comments have been addressed

2. Is the manuscript technically sound, and do the data support the conclusions?

Reviewer #1: (No Response)

Reviewer #2: Yes

3. Has the statistical analysis been performed appropriately and rigorously? 

Reviewer #1: (No Response)

Reviewer #2: Yes

4. Have the authors made all data underlying the findings in their manuscript fully available?

Reviewer #1: (No Response)

Reviewer #2: Yes

5. Is the manuscript presented in an intelligible fashion and written in standard English?

Reviewer #1: (No Response)

Reviewer #2: Yes

6. Review Comments to the Author

Reviewer #1: (No Response)

Reviewer #2: (No Response)

7. PLOS authors have the option to publish the peer review history of their article (what does this mean?). If published, this will include your full peer review and any attached files.

Reviewer #1: No

Reviewer #2: No

---

## [Editor Report · Acceptance letter]

6 Apr 2023

PONE-D-22-29433R1 

Exploring the associations between number of children, multi-partner fertility and risk of obesity at midlife: Findings from the 1970 British Cohort Study (BCS70) 

Dear Dr. Stannard:

I'm pleased to inform you that your manuscript has been deemed suitable for publication in PLOS ONE. Congratulations! Your manuscript is now with our production department. 

Kind regards, 

on behalf of

Dr. Emily W. Harville 

Academic Editor

PLOS ONE